# Using secondary cases to characterize the severity of an emerging or re-emerging infection

Tim K. Tsang [1,2], Can Wang[1], Bingyi Yang[1], Simon Cauchemez [3,4] & Benjamin J. Cowling [1,2,4 ✉]

The methods to ascertain cases of an emerging infectious disease are typically biased toward cases with more severe disease, which can bias the average infection-severity profile. Here, we conducted a systematic review to extract information on disease severity among index cases and secondary cases identified by contact tracing of index cases for COVID-19. We identified 38 studies to extract information on measures of clinical severity. The proportion of index cases with fever was 43% higher than for secondary cases. The proportion of symptomatic, hospitalized, and fatal illnesses among index cases were 12%, 126%, and 179% higher than for secondary cases, respectively. We developed a statistical model to utilize the severity difference, and estimate 55% of index cases were missed in Wuhan, China. Information on disease severity in secondary cases should be less susceptible to ascertainment bias and could inform estimates of disease severity and the proportion of missed index cases.

[1] WHO Collaborating Centre for Infectious Disease Epidemiology and Control, School of Public Health, Li Ka Shing Faculty of Medicine, The University of Hong Kong,, Hong Kong, China. [2] Laboratory of Data Discovery for Health Limited, Hong Kong Science and Technology Park, New Territories, Hong Kong, China. [3] Mathematical Modelling of Infectious Diseases Unit, Institut Pasteur, UMR2000, CNRS, Paris, France. [4]These authors contributed equally: Simon Cauchemez, Benjamin J. Cowling.  ✉email: bcowling@hku.hk

Characterizing the transmissibility and the severity of infectious diseases are two urgent priorities when a new infectious disease emerges, which is highlighted by recent COVID-19 global pandemic[1–3]. While reasonable estimates of transmissibility can usually be inferred from growth rates in identified cases[1,4], it can be more challenging to obtain accurate initial estimates of the clinical severity of infections. Following the influenza A(H1N1)pdm09 pandemic in 2009–10, the Fineberg report identified problems in the measurement of clinical severity as a major challenge to a measured public health response to an influenza pandemic[5]. Early estimates of severity of new infectious diseases need to take into account the potential for ascertainment bias and censoring of outcomes[6–11].

One difficulty in many emerging diseases is the under-ascertainment of all cases, because mild cases may escape clinical detection and hence detected cases may be more severe on average[12]. One possible approach to avoid this selection bias, and obtain improved disease severity, is to focus on the severity of secondary cases that are prospectively identified, such as contact tracing studies with household transmission study as a special case[13]. In these studies, index cases ascertained either by presenting for medical attention or reporting symptoms are not likely to be representative of all infections, because milder cases would have a lower probability of being ascertained. However, the infections among secondary cases may give a more representative picture of the severity of natural cases.

In contact tracing studies, as with volunteer challenge studies, the numbers of cases may not be sufficient to allow observation of the risk of very severe disease following infection, depending on the virulence of the virus. For example, previous systematic review suggested that the case fatality risk (CFR) for influenza A(H1N1)pdm09 ranged from 0.001% to 0.01%[14], but the CFR could be up to 20% for Middle East Respiratory Syndrome (MERS)[10]. Therefore, properly determining the sample size requirement for using this approach would be important. Here, we reviewed the literature on COVID-19 on its severity profile, including spectrum of symptom, severity level and mortality rate among cases, on contact tracing studies that collected information for both index cases and secondary cases. By systematically reviewing and analyzing published data, we aim to characterize the difference in severity for index and secondary cases. We developed a statistical approach to use the severity level of cases to estimate the number of undetected index cases. Finally, we conducted a simulation to determine the sample size requirement of household transmission studies to characterize the severity of emerging infectious diseases.

## Results

In the systematic review, we identified 4068 studies in our search, 885 duplicates were excluded. After screening the titles and abstracts of the remaining articles, 599 full texts were screened (Fig. S1). On the basis of our selection criteria, 561 of those studies were excluded and 38 met our inclusion criteria[15–52] (Table S1). Of these, 19 studies provided information on types of symptoms[15–23,26–29,31,38,40,41,44,48], 10 studies provided information on case severity[15,18,23,24,26,29,30,32,35,44], 33 studies provided symptom status[15,16,18,19,21–26,28–30,32,34–42,44,46–52], 10 studies provided hospitalization[30,33,34,36,37,39,43,48,50,51], and 7 studies provided fatality[27,30,33,37,39,43,45] for both index and secondary cases. Overall, 50,382 index cases and 30,309 secondary cases provided information on at least one measure of clinical severity and type of symptoms. For ascertainment methods of index cases, almost all studies used PCR, with seven studies additionally used either symptom[17,43], radiology[17,26,27,33] or serology[21,27,33,46], while one study did not use PCR and used

rapid test[34]. For ascertainment methods of secondary cases, majority studies used PCR, with seven studies additionally used either symptom[17,43], radiology[17,26,27,33] or serology[22,27,33,46], while three studies used serology[25,39,48] and one study used rapid test[34], but not PCR. Regarding to the test coverage among secondary cases, 23 studies tested all identified contacts to detect secondary cases, 2 studies only tested symptomatic persons[20,45], 4 studies tested less than 70% of identified contacts[34,40,43,50] and 5 studies provided no information on test converge[17,19,21,46,49]. We summarized the extracted information on clinical severity and types of symptoms for index cases and secondary cases (Table 1). Overall, we consistently found that the severity for index cases was higher than for secondary cases.

**Types of symptoms.** In general, the frequency of symptoms for secondary cases was lower than for index cases (Tables 1 and S2). For fever (Fig. 1), 16/18 studies reported higher frequency for index cases than secondary cases[15–17,19–23,26,27,29,31,38,40,41,48], with 11 of them reported significant difference[15,17,19,21–23,26,29,38,41,48]. For cough (Fig. 2), 17/19 studies reported higher frequency for index cases than secondary cases[15–23,26–29,38,41,44,48], with 7 of them reported significant differences[15,22,23,26,29,38,48]. Nine studies[17,21–23,26,29,38,48] reported higher frequency of all recorded symptoms for index cases than for secondary cases, with four of them reaching statistical significance for all recorded symptoms[22,23,26,29]. Overall, the proportion of all symptoms for index cases were 1.09 to 1.88-fold higher than for secondary cases, with statistical significance for all symptoms except for sore throat. The heterogenicity of the risk ratio comparing the proportion of symptoms for index and secondary cases were low except for fever (Table 1). The most commonly reported symptoms were fever and cough. Based on the reported symptom of secondary case, 45.0% (95% confidence interval (CI): 42.9%, 47.1%) and 36.3% (95% CI: 34.4%, 38.3%) of cases had fever and cough respectively. The proportions of fever and cough for index cases were 43% (95% CI: 24%, 66%) and 34% (95% CI: 19%, 50%) higher than for secondary cases respectively.

**Clinical severity.** Among the 10 identified studies reporting case severity (Fig. 3), 9 studies[15,18,23,24,26,29,30,32,35], reported higher case severity for index cases than for secondary cases, with 6 studies[15,24,26,29,30,32] reporting a statistically significant difference (Fig. 3 and Table S3). One study reported significantly lower case severity for index cases than for secondary cases[44]. Overall, based on the reported case severity of secondary cases (Table 1), 13.4% (95% CI: 12.8%, 14.1%) of cases were severe/critical. Index cases were more severe than secondary cases. The proportion of severe/critical in index cases were 72% (95% CI: 6%, 179%) higher than in secondary cases. However, the degree of difference in case severity for index and for secondary cases varied (Table 1).

**Symptom status, hospitalization, and fatality for COVID-19.** Among 31 studies reporting symptom status of cases (Fig. 4), 26 studies reported higher proportion of being symptomatic for index cases than secondary cases[15,16,19,21–26,28–30,32,34–36,38–42,44,46,47,49,51], and 15 studies reached statistical significance[15,19,22,24–26,28–30,32,36,38–40,49]. Overall, based on the secondary cases, 90.8% (95% CI: 90.3%, 91.3%) of cases were symptomatic. The proportion of being symptomatic for index cases was 12% (95% CI: 6%, 18%) higher than for secondary cases. However, the difference in symptomatic proportion for index and for secondary cases varied (Table 1).

Among 10 studies reporting the hospitalization status of cases (Fig. 5A), 8 studies reported higher proportion of hospitalization for index cases than secondary cases[30,33,36,39,43,48,50,51], and 5 studies reached statistical significance[33,43,48,50,51]. Overall, based

**Table 1 Summary of symptoms and clinical severity for COVID-19. 95% confidence intervals for proportion are computed by exact binomial method.**

|  | Index cases | Secondary cases | Risk ratio (95% CI) | p-value | Heterogeneity ($I^2$) |
|---|---|---|---|---|---|
|  | Count; Proportion (95% CI) | Count; Proportion (95% CI) |  |  |  |
| *Symptom* |  |  |  |  |  |
| Fever | 3753/5551; 67.6% (66.4%, 68.8%) | 972/2160; 45% (42.9%, 47.1%) | 1.43 (1.24, 1.66) | <0.001 | 84.29 |
| Cough | 2374/5683; 41.8% (40.5%, 43.1%) | 848/2335; 36.3% (34.4%, 38.3%) | 1.34 (1.19, 1.5) | <0.001 | 61.22 |
| Sore throat | 437/5333; 8.2% (7.5%, 9%) | 233/1884; 12.4% (10.9%, 13.9%) | 1.09 (0.85, 1.4) | 0.502 | 45.97 |
| Headache | 640/5101; 12.5% (11.6%, 13.5%) | 225/1629; 13.8% (12.2%, 15.6%) | 1.32 (1.00, 1.74) | 0.047 | 64.5 |
| Diarrhea | 442/5044; 8.8% (8%, 9.6%) | 137/1771; 7.7% (6.5%, 9.1%) | 1.44 (1.19, 1.74) | <0.001 | 0.01 |
| Fatigue | 1641/5450; 30.1% (28.9%, 31.3%) | 406/2110; 19.2% (17.6%, 21.0%) | 1.64 (1.39, 1.92) | <0.001 | 44.82 |
| Myalgia | 889/5446; 16.3% (15.4%, 17.3%) | 221/1990; 11.1% (9.8%, 12.6%) | 1.88 (1.45, 2.44) | <0.001 | 56.59 |
| *Clinical severity* |  |  |  |  |  |
| *Case severity* |  |  |  |  |  |
| Asymptomatic | 743/31491; 2.4% (2.2%, 2.5%) | 602/11475; 5.2% (4.8%, 5.7%) | 0.39 (0.21, 0.72) | 0.003 | 88.48 |
| Mild | 15400/31491; 48.9% (48.3%, 49.5%) | 5726/11475; 49.9% (49%, 50.8%) | 0.84 (0.69, 1.04) | 0.108 | 75.2 |
| Moderate | 9433/31491; 30% (29.4%, 30.5%) | 3606/11475; 31.4% (30.6%, 32.3%) | 1.08 (0.93, 1.26) | 0.285 | 84.31 |
| Severe/critical | 5914/31491; 18.8% (18.3%, 19.2%) | 1541/11475; 13.4% (12.8%, 14.1%) | 1.72 (1.06, 2.79) | 0.028 | 83.17 |
| Proportion of symptomatic cases | 33692/34760; 96.9% (96.7%, 97.1%) | 12740/14026; 90.8% (90.3%, 91.3%) | 1.12 (1.06, 1.18) | <0.001 | 98.11 |
| Proportion of hospitalization | 354/1908; 18.6% (16.8%, 20.4%) | 121/1759; 6.9% (5.7%, 8.2%) | 2.26 (1.87, 2.74) | <0.001 | 0.01 |
| Proportion of death | 220/16533; 1.3% (1.2%, 1.5%) | 28/5371; 0.5% (0.3%, 0.8%) | 2.79 (1.48, 5.25) | 0.002 | 33.19 |

Risk ratios, their 95% confidence intervals and p-values are computed by random effects meta-analyses with using the inverse variance method and restricted maximum likelihood estimator for heterogeneity. All statistical tests are two-sided tests. Adjustments are not made for multiple comparisons.

on the secondary cases, 6.9% (95% CI: 5.7%, 8.2%) of cases were hospitalized. The proportion of hospitalization for index cases was 126% (95% CI: 87%, 174%) higher than for secondary cases, with low heterogeneity (Table 1).

Among 7 studies reporting fatality status of cases (Fig. 5B), all studies reported higher proportion of fatality cases for index cases than secondary cases[27,30,33,37,39,43,45], and 2 studies reached statistical significance[27,45]. Overall, based on the secondary cases, 0.5% (95% CI: 0.3%, 0.8%) of cases died. The case fatality risk for index cases was 179% (95% CI: 48%, 425%) higher than for secondary cases, with low heterogeneity (Table 1).

**Risk of bias**. Given that the severity of cases could depend on their age for many infectious diseases, including COVID-19, the role of age in the severity difference between index and secondary cases should be explored. If the testing of contacts of all index cases would also depend on age, then age could be a cofounder on the association between severity and being an index case. Hence, we conducted a sensitivity analysis to determine the difference in severity between index and secondary cases, restricted on studies that tested all contacts of index cases, so that the probability of being tested was independent of age (Table S4). For all severity measure, the severity for index cases was significantly higher than for secondary cases. This suggested these severity differences could not be explained by age difference in index and secondary cases.

Next, we tested if the methods to ascertain infections could explain the difference in severity between index and secondary cases. Hence, we conducted a sensitivity analysis to determine the difference in severity between index and secondary cases, restricted on studies that tested all contacts of index cases, and using PCR to confirm infections (Table S5). Only one study fulfilled this criterion for hospitalization and fatality risks and hence these two measures were ignored. For symptoms, we still observed significant higher proportion of fever, cough, fatigue and myalgia for index cases than for secondary cases. For clinical severity, we still observed higher case severity and proportion of being symptomatic for index cases than for secondary cases, and 75.6% (73.9%, 77.1%) of secondary cases were symptomatic.

**Estimation of undetected index cases**. Seven studies reported case severity were conducted in China, with using the 5th case definition in China, defining the case severity to asymptomatic, mild, moderate, severe and critical (Supplementary Note 1). It should be noted that in Luo et al.[29], only index cases with at least one secondary case were reported. In Li et al.[26], clinically confirmed and laboratory-confirmed cases were mixed, and also testing on asymptomatic contacts of cases started February 23, 2020. In Hu et al.[23], only clusters with child cases were included in the study. In Bi et al.[15], the severity of cases was determined by the first clinical assessment of cases, not the whole illness episode. We used the secondary cases in Luo et al.[29], and Hu et al.[24] to estimate the distribution of case severity, since the secondary cases were laboratory-confirmed and the case severity was determined by whole illness episode (Table 2). We found the estimated distributions for these two studies were different, which may be due to the evolving definitions of severity due to study period, or geographical difference.

Then, we estimated the number of undetected index cases for Guangzhou and Wuhan, China based on the observed number of index cases in Luo et al.[29] and Li et al.[26] (Table 2), since these two studies aimed to identify all cases in their study regions. In Luo et al.[29], 147 index cases were ignored since they were imported cases. The severity information of 68/244 (28%) of index cases from January 13 to March 6, 2020 were available. After imputing the missing information for index cases and using the distribution

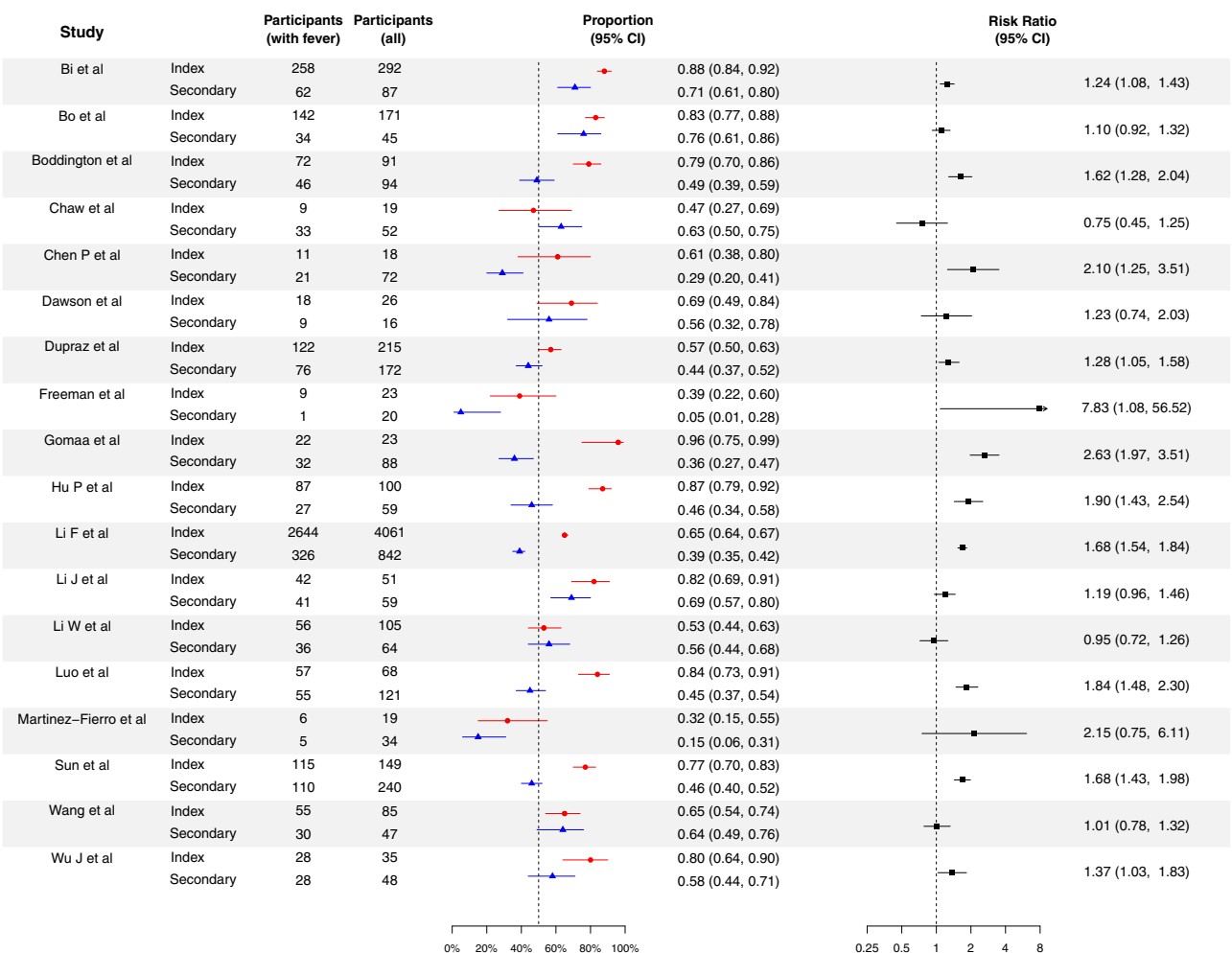

**Fig. 1 Proportion of cases with fever for the index cases and secondary cases for COVID-19, and their corresponding risk ratio of index cases, compared with secondary cases.** Red circles and lines indicate the proportion of fever of index cases and the corresponding 95% exact binomial confidence intervals. Blue triangles and lines indicate the proportion of fever of secondary cases and the corresponding 95% confidence intervals. Black squares and lines indicate the risk ratio of proportion of fever of index cases compared with secondary cases and the corresponding 95% confidence intervals calculated based on normal approximation. All statistical tests are two-sided tests. Adjustments are not made for multiple comparisons.

of case severity based on the secondary cases in the same study, we estimated that there should be 830 (95% credible interval (CrI): 421, 1539) index cases, suggesting that 67% (95% CrI: 42%, 84%) of index cases were missed due to ascertainment bias. In Li et al.[26], the severity information of 29578/38563 (77%) of index cases from December 2, 2019 to April 18, 2020 were available. We used the distribution of case severity based on the secondary cases in Hu et al.[24], which were based in Hunan and included Wuhan, due to the abovementioned limitation in the study design in Li et al.[26]. After imputing the missing information for index cases, we estimated that there should be 88,349 (95% CrI: 65,545, 119,106) index cases, suggesting that 55% (95% CrI: 41%, 68%) of index cases were missed due to ascertainment bias. In both studies, >95% of asymptomatic index cases was missed. Based on the 97 paired index and secondary cases in Maltezou et al. and Xie et al., we estimated that the odds ratio of being moderate or severe secondary cases, for the corresponding index cases were moderate or severe, was 1.29 (95% CI: 0.42, 3.93), compared with corresponding index cases were mild or asymptomatic.

**Sample size requirement of estimating severity.** We conducted a simulation based on household transmission model to determine the sample size requirement of obtaining the upper bound or

lower bound of CFR (as an example of measure of severity). In the simulation we assumed that all cases would be ascertained regardless of symptoms. In general, when the transmission probability doubled, the required number of households could be halved to obtain the same precision of estimates (Fig. 6). For disease with lower severity (such as less lethal strain for pandemic influenza) at 0.1% CFR, we found that around 580 households were needed in order to get the upper bound of the estimate to 10-fold of the corresponding true values for transmissible strain with high transmissibility (20% transmission probability). For the disease with moderate or high severity and high transmissibility at 1% or 10% CFR, respectively 940 and 100 households were needed to obtain the upper bound of estimate to 2-fold of the true values and 1380 and 140 households were needed to obtain the lower bound of estimate to be half of the true values. We also conducted a simulation study assuming there were 50% children and 50% adults in households, and the CFR for children was half for adults (Fig. S2). Overall, slightly more households would be needed since the sample size requirement would depend on precision for estimates of CFRs for both children and adults. For example, for disease with 5% CFR for children and 10% for adults with 20% transmission probability, 360 households would be needed to obtain the upper bound of both estimates for CFRs for children and adults to 2-fold of the true values.

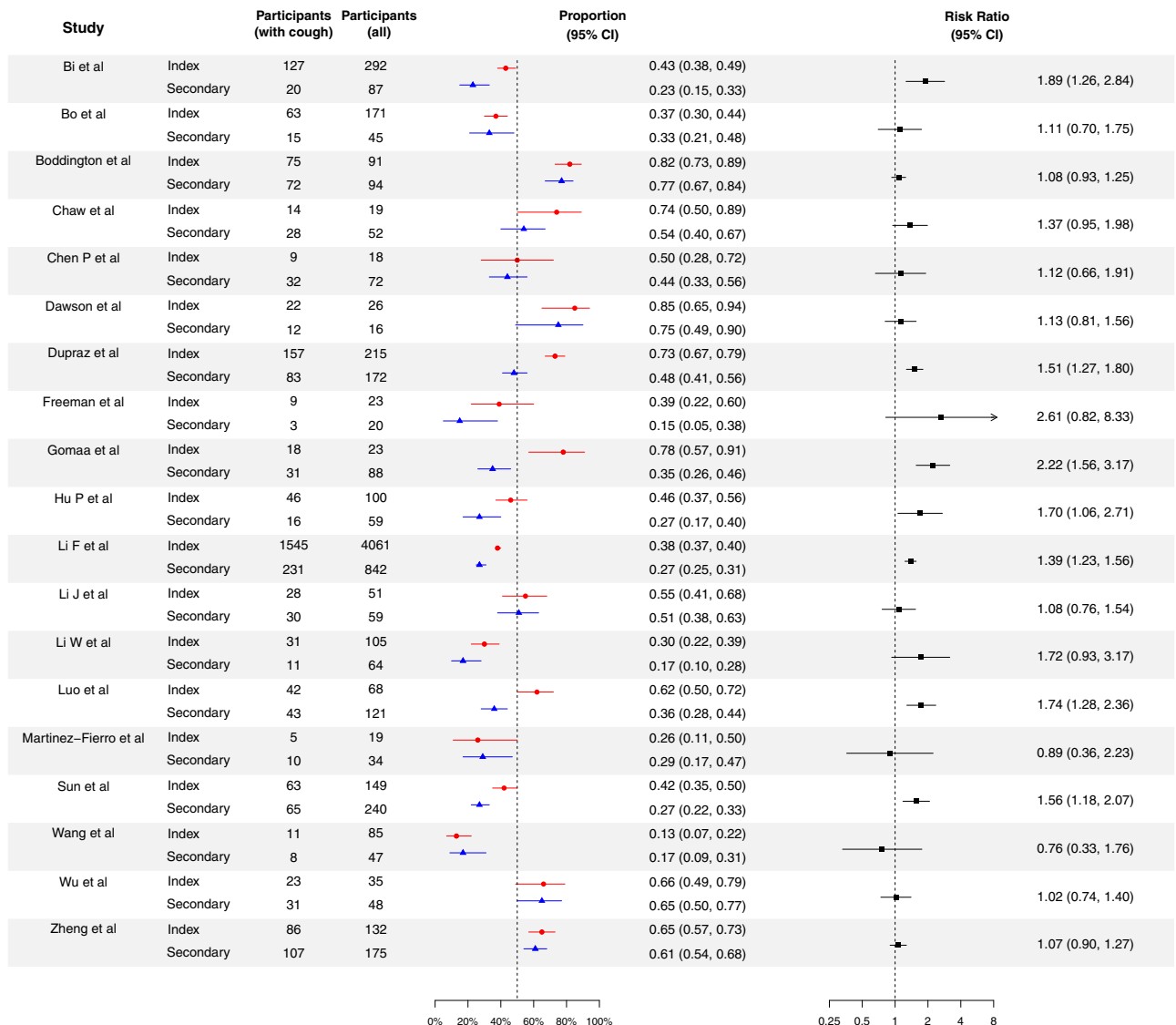

**Fig. 2 Proportion of cases with cough for the index cases and secondary cases for COVID-19, and their corresponding risk ratio of index cases, compared with secondary cases.** Red circles and lines indicate the proportion of fever of index cases and the corresponding 95% exact binomial confidence intervals. Blue triangles and lines indicate the proportion of fever of secondary cases and the corresponding 95% confidence intervals. Black squares and lines indicate the risk ratio of proportion of fever of index cases compared with secondary cases and the corresponding 95% confidence intervals calculated based on normal approximation. All statistical tests are two-sided tests. Adjustments are not made for multiple comparisons.

## Discussion

In this study, we assessed how the presence of different types of symptoms, case severity, symptom status, hospitalization status and fatality status could differ between index and secondary cases in contact tracing studies for COVID-19. For these measures of severity, we found that index cases were generally more severe secondary cases. This confirms that the severity profile of index cases is biased toward infections that cause more serious illness than average. In general, the heterogeneity of the risk ratio comparing severity of index cases and secondary cases were low ($I^2 < 75\%$), suggesting that this ascertainment bias was consistent across studies.

This ascertainment bias is due to way index cases are identified, since cases need to present for medical attention with particular symptoms in order to be identified as a case. For COVID-19, identification of index cases may be conducted on fever[44] or emergency clinics[25], passive surveillance in hospitals[24,42], or presence of symptoms in studies[22]. In some periods, testing may also be focused on symptomatic individuals only due to limited

supply of tests[20,45]. For other infectious disease like influenza (Supplementary Note 2 and Table S6), cases were required to have a certain number of symptoms, such as cough or fever, in order to be enrolled as index cases in household transmission studies. Therefore, the proportion of index cases with cough could be >90% for some household studies[53–55]. In contact tracing studies for MERS (Supplementary Note 2 and Fig. S3), the case with the most severe outcome in a cluster was likely to be the index case. Therefore, the CFR could be >90% in those studies[56,57]. Also, the risk of ICU admission for secondary cases were around half that of index cases in a study of the avian influenza A(H7N9) virus[58]. However, for some very severe diseases, such as avian influenza A(H5N1) virus with >50% CFR, this bias may be less important[58]. Other measures of severity may also be affected, for example the viral loads and the duration of viral shedding of influenza[59].

It should be noted that only a fraction of identified close contacts are ascertained in some studies, which may cause bias in estimating the distribution of case severity. For example, when

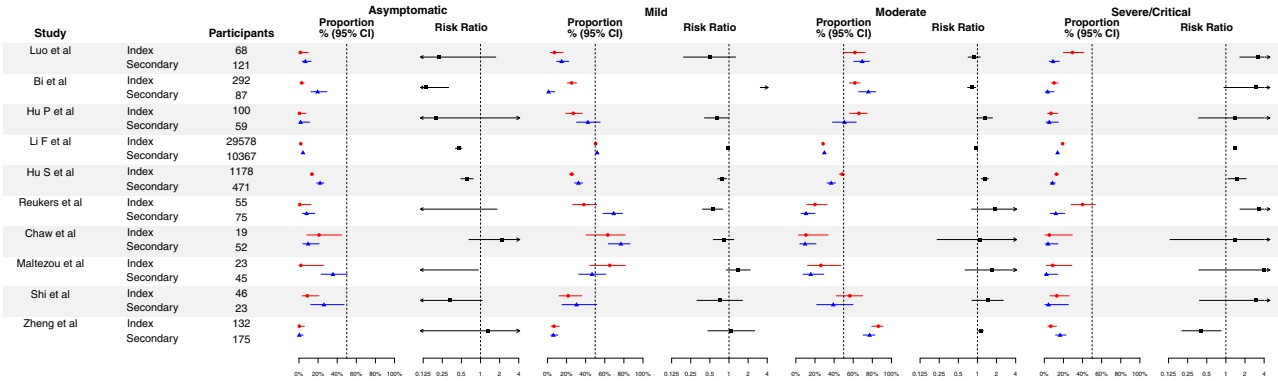

**Fig. 3 Reported case severity, including proportion of asymptomatic, mild, moderate and severe of critical for COVID-19 for index cases and secondary cases, and their corresponding risk ratio of index cases verse secondary cases.** Red circles and lines indicate the proportion of fever of index cases and the corresponding 95% exact binomial confidence intervals. Blue triangles and lines indicate the proportion of fever of secondary cases and the corresponding 95% confidence intervals. Black squares and lines indicate the risk ratio of proportion of fever of index cases compared with secondary cases and the corresponding 95% confidence intervals calculated based on normal approximation. All statistical tests are two-sided tests. Adjustments are not made for multiple comparisons.

only symptomatic close contacts were tested, so that asymptomatic cases could not be identified[20,35,43,45,60]. While this practice can save resources, it leads to under-ascertainment of asymptomatic infections. In previous systematic reviews, the asymptomatic fraction was estimated to be up to 4%–28% for influenza[61], 27% (95% CI: 14.5%–39.6%) for ebola[62]. For pertussis, 56% of tested asymptomatic contacts were laboratory-confirmed cases[63]. Accounting for asymptomatic cases would be important to characterizing the disease severity. Owing to the potential of pre-symptomatic and asymptomatic transmission for COVID-19[52,64,65], testing on asymptomatic persons became common practice in contact tracing studies of COVID-19, which would address the ascertainment bias for asymptomatic cases.

Therefore, to reduce the ascertainment bias in estimating severity of diseases, It is recommended to use the severity profile in secondary cases, in studies that test all close contacts regardless of the presence of illness[10,12,66]. Such approach allows the detection of asymptomatic infections, leading to a more accurate assessment of severity of disease.

Based on the secondary cases, we estimated the proportion of cases with fever and cough were 45% and 36% respectively. Eight previous systematic reviews[67–74] did not stratify their analysis based on the approach for detecting infections (Table S7), and all found that the proportional of fever and cough were higher than our estimates, even though four systematic reviews restricted their analysis to children only, in which the severity of COVID-19 tends to be lower compared with adults[70–73], which was also consistent with the ascertainment bias. We estimated 91% of cases were symptomatic, which was consistent with a previous systematic review suggested that 9% of cases were asymptomatic[75]. However, these estimates could depend on age group, with estimates symptomatic proportion for children was generally lower[70,71,73,75] (Table S7), or converge of test, as the estimates was 73% when restricted to studies with PCR-confirmed infection and testing on all contacts. On the other hand, the estimates of symptomatic proportion based on serology data would be much lower, range from 25% and 2–13% for two systematic reviews[76,77]. This difference is also observed in influenza, where the asymptomatic fraction could be much higher for serology studies compared with case-ascertained studies[61]. One potential reason was the imperfect timing of collection of swabs, resulting in reduced sensitivity to detect infections[78]. Therefore, some infections may be missed by PCR but could be captured by serology[79,80]. In this case, severity measure may be overestimated

if using PCR-confirmed infections. On the other hand, the estimate from serology could mistakenly classify uninfected persons as infected because of cross reactions in serology, particularly when only convalescent sera are collected[81–85]. Further research on the discrepancy on symptomatic proportion based on serology and PCR would be required.

We estimated that the CFR was 0.5%, which was consistent with the estimates from a systematic review[86] and some estimates from the surveillance case count[87]. However, the CFR would likely depend on age[88], or potential geographical differences such as availability of medical resources[3], or could be higher for high-risk groups such as medical personnel[89]. Some previous estimates could be up to 2–4% in the early stage of pandemic based on Wuhan, which was also consistent with the ascertainment bias that during the pandemic in Wuhan, many mild cases were undetected and hence the CFR was overestimated[90].

The proportion of symptomatic cases for index cases was 50% higher than for secondary cases, suggesting that a portion of cases were undetected. By using the distribution of case severity estimated from the laboratory-confirmed secondary cases in Guangzhou and Hunan, China, we estimated that 68% and 56% of index cases were undetected in Guangzhou and Wuhan, China respectively. It was impossible to estimate all undetected cases, because some of them could be secondary cases for more than one index cases, depending on the unobserved transmission networks. Our estimates of undetected index cases were consistent with previous estimates using different methods, such as the modeling based on mobility data[91], changing case definitions[92] or seroprevalence study[93]. The observed severity of COVID-19 among secondary cases was still heterogeneous, likely due to changing case definition when the pandemic evolved[92], or limited resource during the pandemic outbreak so that the intensity of contact tracing changed. It should be noted that in our analysis it is assumed that the severity distribution of index and secondary cases were independent. Although it was supported by two studies identified in our systematic reviews, it was also biological plausible that the severity of index and secondary cases were correlated. Also, we assumed that all secondary cases were identified, which was also reasonable since the ascertainment bias still exists when focusing on studies tested all identified. However, if these assumptions were invalid, we would expect that the severity for secondary cases would be overestimated, because clusters with both mild index and secondary cases would be more likely to be missed. Therefore, the number of missed index cases

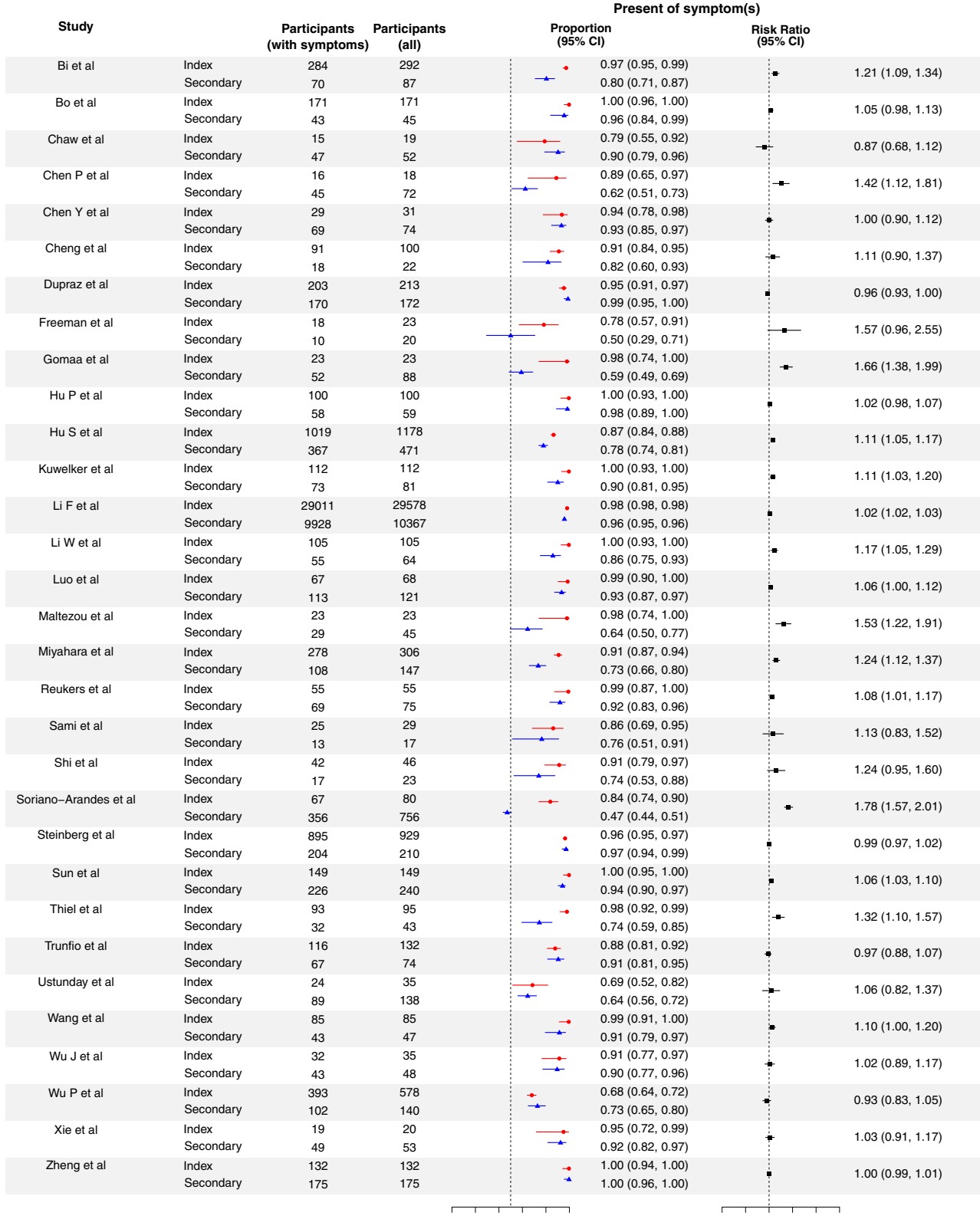

**Fig. 4 Proportion of symptomatic cases for the index cases and secondary cases for COVID-19, and their corresponding risk ratio of index cases, compared with secondary cases.** Red circles and lines indicate the proportion of fever of index cases and the corresponding 95% exact binomial confidence intervals. Blue triangles and lines indicate the proportion of fever of secondary cases and the corresponding 95% confidence intervals. Black squares and lines indicate the risk ratio of proportion of fever of index cases compared with secondary cases and the corresponding 95% confidence intervals calculated based on normal approximation. All statistical tests are two-sided tests. Adjustments are not made for multiple comparisons.

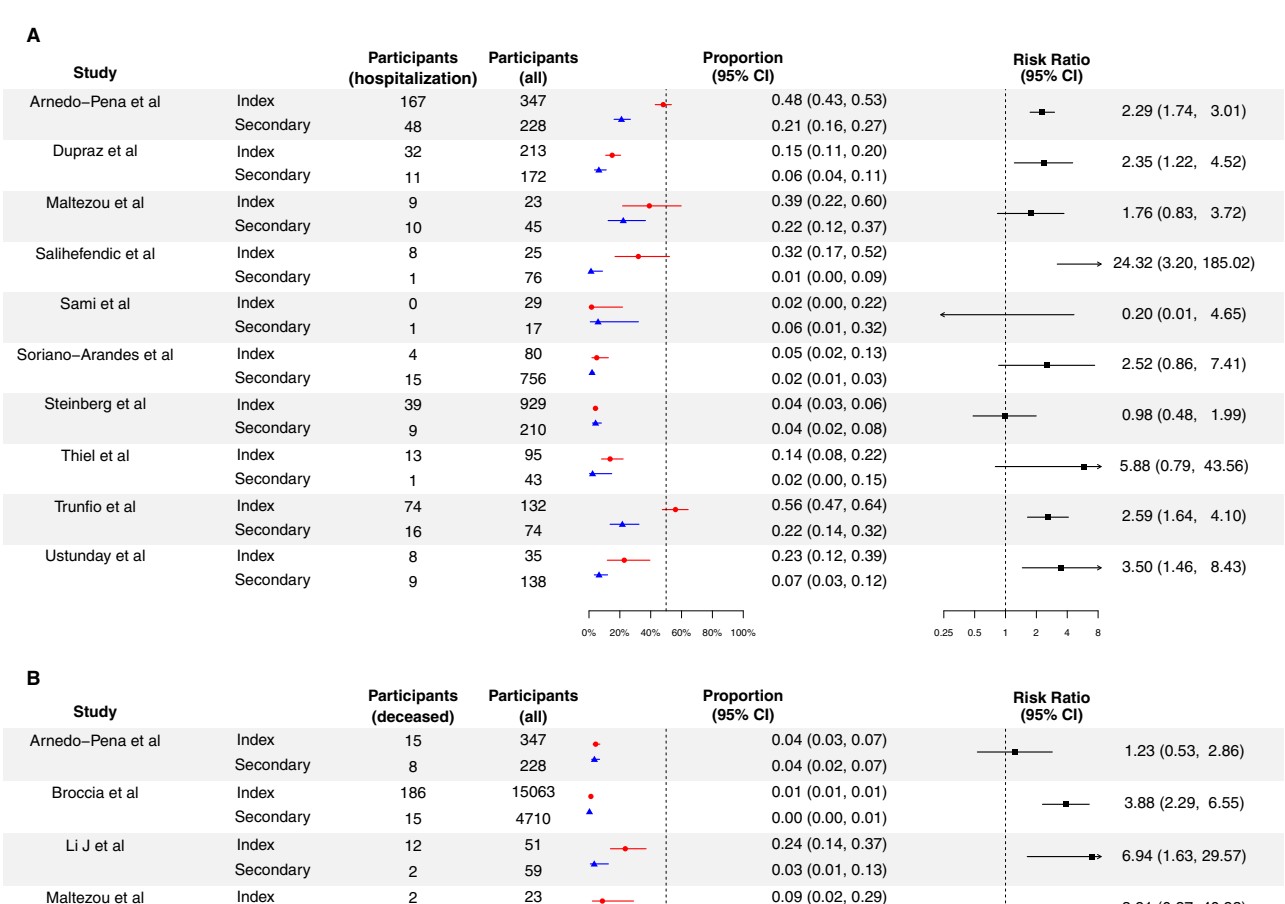

**Fig. 5 Proportion of cases with hospitalization and death for the index cases and secondary cases for COVID-19, and their corresponding risk ratio of index cases, compared with secondary cases.** Proportion of cases with hospitalization (**A**) and death (**B**) for the index cases and secondary cases for COVID-19, and their corresponding risk ratio of index cases, compared with secondary cases. Panel A and B indicate the proportion of hospitalization and death respectively. Red circles and lines indicate the proportion of fever of index cases and the corresponding 95% exact binomial confidence intervals. Blue triangles and lines indicate the proportion of fever of secondary cases and the corresponding 95% confidence intervals. Black squares and lines indicate the risk ratio of proportion of fever of index cases compared with secondary cases and the corresponding 95% confidence intervals calculated based on normal approximation. All statistical tests are two-sided tests. Adjustments are not made for multiple comparisons.

due to case ascertainment bias would be underestimated, and our estimates of total number of index cases would be a lower bound of the actual number of index cases.

For disease with high severity such as MERS, the CFR could be estimated from the analysis of secondary cases in contact tracing studies. However, this would be more difficult to do for diseases with low severity such as H1N1pdm09[14]. Indeed, in such scenario, our simulation study suggests that 1000+ households may be needed. For a disease with low severity, other settings to identify secondary cases may be needed, such as contact tracing for cases identified based on routine surveillance[10,15]. We also explored how different severity levels among groups might impact performance and found that the sample size requirement would be slightly higher to estimate CFRs in multiple groups.

There could be other potential explanations for the difference in severity between index and secondary cases. While there could be difference in age distribution between index and secondary cases and hence age may explain this difference, our sensitivity analysis that only included those studies tested all contacts of index cases suggested that this severity difference still exist. Also, a study compared pediatric index cases and secondary cases in the same region[36], and found that the proportion of being symptomatic for index cases were 78% (95% CI: 57%, 101%) higher than secondary cases. Given that index cases were identified before secondary cases, and therefore the treatments may be improved, and hence secondary cases may get better treatments, compared with index cases. However, this was unlikely to completely explain this difference, since the only drug treatments approved by U.S. Food and Drug Administration on October 2020[94], and the identified studies spanned the period from January to December 2020.

Our study has a number of limitations. First, since the index cases were more severe compared with average cases, if the severity of secondary cases was affected by the characteristics of

**Table 2 Observed number of secondary cases by reported severity level for COVID-19 in China, and the estimated distribution of severity levels of natural infections based on secondary cases.**

| | Asymptomatic | Mild | Moderate | Severe or critical | Total |
|---|---|---|---|---|---|
| *Estimated distribution of severity levels of natural infections* | | | | | |
| *Luo et al. (Guangzhou, China)* | | | | | |
| Number of secondary cases | 8 | 18 | 84 | 11 | 121 |
| Estimated distribution of severity level | 0.07 (0.03, 0.12) | 0.15 (0.09, 0.22) | 0.69 (0.6, 0.77) | 0.09 (0.05, 0.15) | |
| *Hu et al. (Hunan, China)* | | | | | |
| Number of secondary cases | 104 | 153 | 174 | 40 | 471 |
| Estimated distribution of severity level | 0.22 (0.18, 0.26) | 0.32 (0.28, 0.37) | 0.37 (0.33, 0.41) | 0.09 (0.06, 0.11) | |
| *Estimated numbers of total index cases before and after correcting for case ascertainment bias* | | | | | |
| *Guangzhou, China from Jan 13 to Mar 6, 2020, severity information available for 68/244 (28%) of index cases (based on Luo et al.)* | | | | | |
| Observed number | 1 | 5 | 42 | 20 | 68 |
| Estimated number before correcting bias | 4 (1, 14) | 18 (8, 34) | 149 (125, 172) | 71 (51, 95) | 244 |
| Estimated number after correcting bias | 51 (19, 131) | 115 (51, 265) | 535 (270, 1108) | 71 (51, 95) | 830 (421, 1539) |
| Proportion of missed index cases | 0.93 (0.59, 0.99) | 0.85 (0.54, 0.95) | 0.72 (0.41, 0.88) | 0 (0, 0) | 0.67 (0.42, 0.84) |
| *Wuhan, China from Dec 2, 2019 to Apr 18, 2020, severity information available for 29578/38563 (77%) of index cases (based on Li et al.)* | | | | | |
| Observed number | 567 | 14928 | 8416 | 5667 | 29578 |
| Estimated number before correcting bias | 739 (711, 769) | 19463 (19360, 19570) | 10972 (10878, 11065) | 7388 (7303, 7474) | 38563 |
| Estimated number after correcting bias[a] | 19236 (13486, 27854) | 28275 (20055, 40448) | 31996 (23059, 45905) | 7388 (7303, 7474) | 88349 (65545, 119106) |
| Proportion of missed index cases | 0.96 (0.94, 0.97) | 0.31 (0.03, 0.52) | 0.66 (0.52, 0.76) | 0 (0, 0) | 0.55 (0.41, 0.68) |

Observed number of index cases by reported severity level for COVID-19 in China, and the estimated numbers of total index cases before and after correcting for case ascertainment bias, assuming all severe/critical cases were detected.
[a]The distribution of severity distribution was estimated based on data from Hu et al.

their infectors, then the severity profile of secondary cases may be biased and may not be representative of all natural infections. Also, the difference in severity between index and secondary cases would be underestimated. The underestimation would be even more severe, if transmissibility and severity were positively correlated. Second, the recruitment methods among studies were different. This may affect the comparability of the results, although all index cases were laboratory-confirmed in almost all studies. Third, the definition of presence of symptoms, and also severity could be different among studies, which may affect the interpretation of results. Forth, other demographic difference between index cases and secondary cases may be able to explain difference in severity, such as index cases had specific occupations and we may not be able to explore for this. Finally, for many diseases including COVID-19, severity is likely related to age[13,95–97], but publicly available data are not always sufficient information to account for this.

In conclusion, we observed higher severity in index cases than in secondary cases in contact tracing studies for COVID-19, consistent with ascertainment bias toward more severe cases. We demonstrated the use of secondary cases in studies testing all close contacts regardless of illness, to estimate the severity profile. We developed a statistical model to estimate the number of undetected index cases, and provided guidelines on the requirement of number of households to estimate disease severity from secondary cases.

## Methods

**Definition of index and secondary cases.** In this study, an index case was defined as the first detected case in a cluster, while secondary cases were defined as the identified cases due to contact tracing triggered by detection of the index case. Some studies may also provide data on sporadic cases, defined as cases without any identified infected contacts, these sporadic cases were also considered as index cases in our analysis. It should be noted that some studies used the term 'primary' case to define the first detected case in a cluster. We excluded index cases that were

travelers, given that they may be substantially different from local people in the study regions.

**Search strategy and selection criteria.** This systematic review was conducted following the Preferred Reporting Items for Systematic Review and Meta-analysis (PRISMA) statement[98]. A standardized search was done in PubMed, Embase and Web of Science, using the search term "((SARS-CoV-2 OR COVID-19) AND (household OR close contact) AND (symptom OR severity OR death OR fatality))". The search was done on 2 September, 2021, with no language restrictions. Additional relevant articles from the reference sections were also reviewed.

Contact tracing studies with at least 15 index cases, and reported types of symptoms and clinical severity among index cases and secondary cases for COVID-19 were included, and the numbers of index cases and secondary cases, were extracted. The following symptoms were included: fever, cough, fatigue, diarrhea, sore throat, headache, and myalgia. Clinical severity was measured as follows: (1) case severity, defined as the following four categories, asymptomatic, mild, moderate and severe/critical, (2) symptom status of cases, (3) hospitalization, and (4) fatality. For each severity measure, we extracted the number of index and secondary cases with or without presence of that severity measure. Individual data with information on severity measures was also extracted if available. We also extracted the age and sex distribution for index and secondary cases, the study period, the coverage of tests of identified contacts, the case ascertainment methods, and the settings of contacts. Studies without severity measures for both index and secondary cases were excluded.

Two authors (T.K.T. and C.W.) independently screened the titles, with disagreement resolved by consensus together with a third author (B.Y.). Studies identified from different databases were de-duplicated. Two authors (T.K.T. and C.W.) independently extracted data from the included studies, with disagreement resolved by consensus with a third author (B.Y.). Our study aimed to summarize the severity difference between index and secondary cases caused by ascertainment bias and therefore this bias would exist by the design of this systematic review. Age distributions of index cases and secondary cases, sampling methods (the test converge of contacts), and the methods to ascertain infections (type of laboratory tests or symptom-based ascertainment) was used as proxies of risk of bias for individual studies[99].

**Data analysis.** For each symptom and severity measures, we computed the risk of the event of interest for index and secondary cases, and the corresponding relative risk for index cases, compared with secondary cases by Fisher's exact tests. We conducted random effects meta-analyses, using the inverse variance method and restricted maximum likelihood estimator for heterogeneity, to summarize the risk

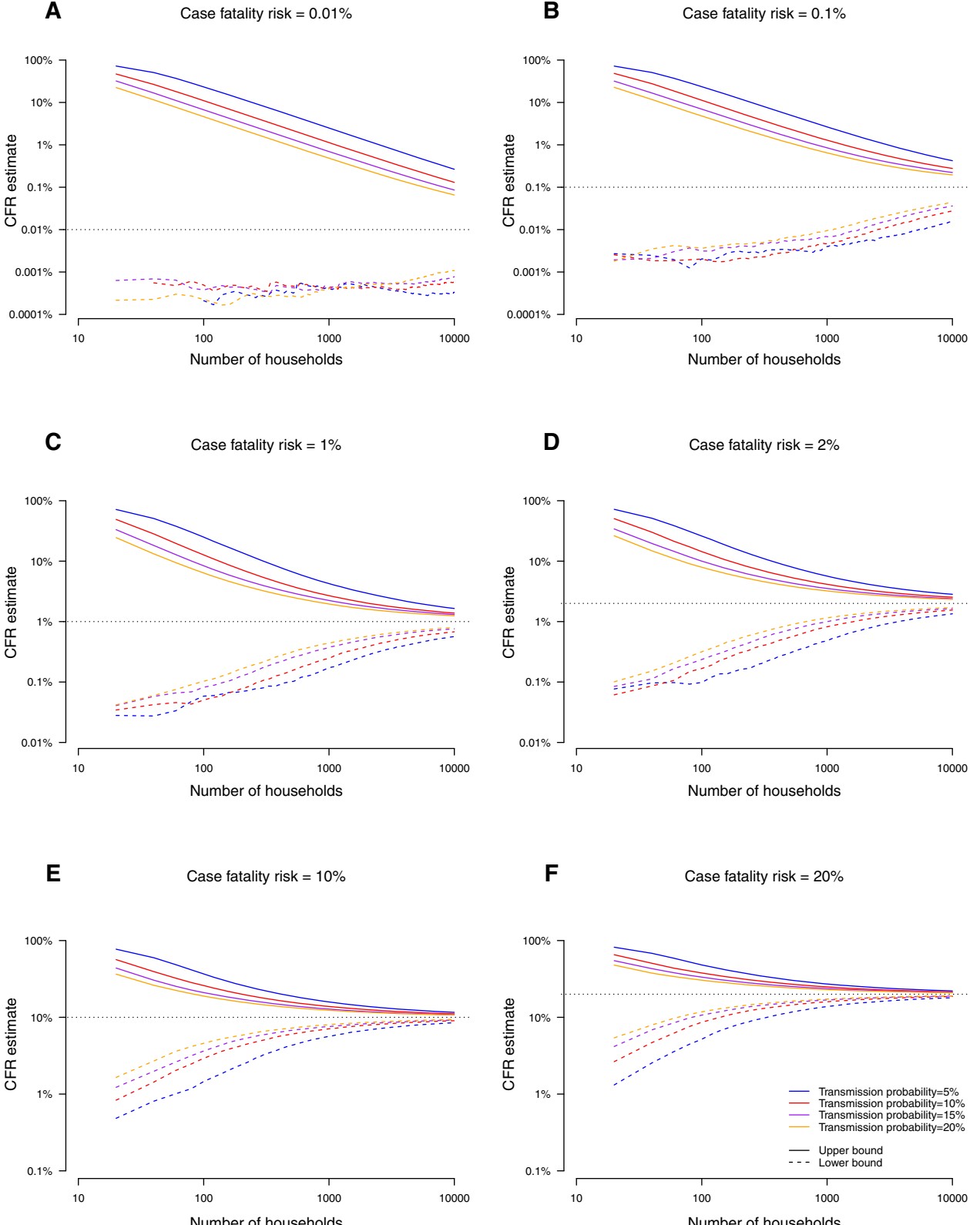

**Fig. 6 The estimates of CFR and associated uncertainy with different number of households.** The lower bound and upper bound of estimates for case fatality risk (CFR) under different value of CFR (Panel A-F), and different secondary infection risk in households.

ratio among the studies for different severity measures[100–103]. Cochran Q test and the $I^2$ statistic were used to identify and quantify heterogeneity among included studies[104,105]. An $I^2$ value >75% indicates high heterogeneity[105].

We aim to use the difference in case severity between index and secondary cases, to determine the number of undetected index cases. We developed a statistical approach to estimate the number of undetected index cases in studies that aimed to identify all cases in a region (Supplementary Note 3). In this approach, we first used a multinomial model to estimate the proportion of each level of case severity after infection (Supplementary Note 3), assuming that all secondary cases were identified and hence the case severity distribution of secondary cases was the same as that of natural infection, and the severity distribution of index cases and secondary cases were independent. After that, by assuming that all severe/critical index cases were observed, we can use the estimated case severity distribution of natural infection to estimate the number of undetected index cases with the other three levels of case severity.

Using the CFR as an example, we conducted a simulation study to determine the power and sample size requirement to estimate the CFR from secondary cases (Supplementary Note 4). We used an individual-based household transmission model[106] to simulate the number of secondary cases and their fatality outcome, with different assumptions on CFR, transmission probability and assumed infectiousness profile since infection (Table S7). We further used the model to determine the impact of the difference in CFR in different groups (children vs adults as an example) on requirement of sample size. We reported the upper bound and lower bound of the estimates of CFR based on different number of households (from 20-10000), to determine the required sample size to get lower bound and upper bound estimates for CFR for a disease. We conducted simulations with different values of CFR for different disease severity, and different values of transmission probability for different disease transmissibility.

**Role of the funding source**. The funder of the study had no role in study design, data collection, data analysis, data interpretation, or writing of the report.

**Reporting summary**. Further information on research design is available in the Nature Research Reporting Summary linked to this article.

## Data availability

All the extracted data in the systematic review is available in the main text and supplementary material. They are also used as input in the modeling analysis.

## Code availability

Statistical analyses were conducted using R version 4.0.5 (R Foundation for Statistical Computing, Vienna, Austria). Code is available at Github: https://github.com/timktsang/severity_review.

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

## Acknowledgements

This project was supported by the Health and Medical Research Fund, Food and Health Bureau, Government of the Hong Kong Special Administrative Region (grant no. COVID190118; B.J.C.) and the Collaborative Research Fund (Project No. C7123-20G; B.J.C.) of the Research Grants Council of the Hong Kong SAR Government. BJC is supported by the AIR@innoHK program of the Innovation and Technology Commission of the Hong Kong SAR Government.

## Author contributions

Study design: T.K.T., S.C., B.J.C. Data collection: T.K.T., C.W., B.Y. Data analysis: T.K.T., C.W., B.Y. Data interpretation: T.K.T., C.W., B.Y., S.C., B.J.C. Wrote first draft: T.K.T. All authors contributed to the final draft.

## Competing interests

B.J.C. reports honoraria from Sanofi Pasteur, GSK, Moderna, and Roche. The authors declare no competing interests.
