## [Peer Review File · Nature Communications]

REVIEWER COMMENTS

Reviewer #1 (Remarks to the Author):

The authors conducted systematic review to characterize difference in severity between index cases and secondary cases in COVID-19 infections. And they attempted to develop a statistical model to estimate the true number of index cases.

The literature review does offer valuable findings by suggesting some systematic difference in symptoms, hospitalization rate and mortality rate. However, I have a few reservations about the novelty and rigorousness of their proposed model. First of all, the model they developed appears to be overly simplified and has to rely on a few strong assumptions. For example, they used two multinomial distributions to describe the "severity distribution" of index cases and secondary cases. It is not clear how they fit the model and how the model describes the relation between index and secondary cases. Also, the assumption of "severe profile of secondary cases is the true severity profile for a disease" is questionable to me. And do they assume secondary cases are fully observed? (which is also questionable if that is the assumption). Finally, it is not clear what is the added novelty given that in the literature there are already approaches estimating undetected cases.

Reviewer #2 (Remarks to the Author):

This manuscript develops a method to improve the estimation of the severity profile of an infectious disease based on secondary cases. This particularly relevant in the context of emerging infectious disease, where information is initially very limited. The method is applied retrospectively to COVID-19.

The paper is relevant and relatively well written. I have a few general comments related to the content and few specific ones related to the form.

1- Typically, this kind of meta-analysis derives indicator such as combined risk ratios based on mixed effect model. Could the authors derive such pooled estimates from their framework and give indication about the heterogeneity between studies. Typically meta-analyses report such pooled value as well as an I-squared value, which reflects (I believe) the heterogeneity between study.

2- While the method seem to give sensible estimates of severity, a lot of new information has been accumulated about COVID-19 severity by now. It seems a missed opportunity not to compare the results from this manuscript to more results from the literature. The submitted manuscript provides a promising method that could be used in real-time during the next emerging disease event. But retrospectively, a lot of detailed study of COVID-19 have taken place, notably including serological-based evidence of severity profile. So how does the proposed method-estimates compared to what has been published? E.g. more systematic comparison?

Where does the figure for MERS-CoV fits in?

Are the figure for death reflecting an CFR or IFR?

Line 83: should read 'by systematically reviewing and analysing'?

Line 109 and throughout: should read 'coverage'?

Line 110: should delete 'except for'?

Line 127: should read 'reaching statistical...'?

Line 149 and other paragraphs below should read 'studies reporting'?

Reviewer #1 (Remarks to the Author):

The authors conducted systematic review to characterize difference in severity between index cases and secondary cases in COVID-19 infections. And they attempted to develop a statistical model to estimate the true number of index cases. The literature review does offer valuable findings by suggesting some systematic difference in symptoms, hospitalization rate and mortality rate.

Response 1.1: Thank you. We have also updated the review to include articles published by September 2, 2021, modifying the text accordingly. The overall conclusions remain the same.

However, I have a few reservations about the novelty and rigorousness of their proposed model. First of all, the model they developed appears to be overly simplified and has to rely on a few strong assumptions. For example, they used two multinomial distributions to describe the “severity distribution” of index cases and secondary cases. It is not clear how they fit the model and how the model describes the relation between index and secondary cases. Also, the assumption of “severe profile of secondary cases is the true severity profile for a disease” is questionable to me. And do they assume secondary cases are fully observed? (which is also questionable if that is the assumption).

Response 1.2: Thanks for your comments. As mentioned by the reviewer, we fitted two multinomial distributions for the ‘severity distribution’, one is for index cases and another one is the secondary cases. Hence, we assumed that the ‘severity distribution’ for index cases and secondary cases are independent. In our systematic review, we identified two studies reported the severity information for paired index cases and secondary cases (Maltezou et al and Xie et al). Based on Maltezou et al. [1] and Xie et al. [2] with 97 index-secondary pairs, we estimated that the odds ratio of being moderate or severe secondary cases, for the corresponding index cases were moderate or severe, was 1.29 (95% CI: 0.42, 3.93), compared with index cases that were mild or asymptomatic, supporting our assumption on the independence of severity distribution for index and secondary cases.

On the other hand, if the severity of index cases and secondary cases were positively correlated, we would expect that the severity for secondary cases would be overestimated, because clusters with both mild index and secondary cases would be more likely to be missed. Therefore, the number of missed index cases due to case ascertainment bias would be underestimated, and our estimates would be a lower bound of the actual number of index cases.

Regarding to the assumption of ‘severe profile of secondary cases is the true severity profile for a disease’ and ‘secondary cases are fully observed”, we believed it is reasonable in studies that tested all identified contacts and hence all secondary cases should be identified. If this is not correct, for example some contacts and hence secondary cases were unidentified, we believe that the estimated severe distribution by our approach would be an overestimation of the true severity, as explained in the previous paragraph. And therefore, the impact on our estimated number of missed index cases would be the same and hence our estimated number of total index cases would be a lower bound of the actual one. We have clarified this in the methods, results, and discussion section in the revised manuscript.

In the result section, we added ‘Based on the 97 paired index and secondary cases in Maltezou et al. and Xie et al., we estimated that the odds ratio of being moderate or severe secondary cases, for the corresponding index cases were moderate or severe, was 1.29 (95% CI: 0.42, 3.93), compared with corresponding index cases were mild or asymptomatic.’

In the discussion section, we added ‘It should be noted that in our analysis it is assumed that the severity distribution of index and secondary cases were independent. Although it was supported by two studies identified in our systematic reviews, it was also biological plausible that the severity of index and secondary cases were correlated. Also, we assumed that all secondary cases were identified, which was also reasonable since the ascertainment bias still exists when focusing on studies tested all identified. However, if these assumptions were invalid, we would expect that the severity for secondary cases would be

overestimated, because clusters with both mild index and secondary cases would be more likely to be missed. Therefore, the number of missed index cases due to case ascertainment bias would be underestimated, and our estimates of total number of index cases would be a lower bound of the actual number of index cases.'

In the methods section, we added 'In this approach, we first used a multinomial model to estimate the proportion of each level of case severity after infection (Appendix), assuming that all secondary cases were identified and hence the case severity distribution of secondary cases was the same as that of natural infection, and the severity distribution of index cases and secondary cases were independent.'

Finally, it is not clear what is the added novelty given that in the literature there are already approaches estimating undetected cases.

Response 1.3: We agree that there are approaches to estimate undetected cases such as changing case definitions and seroprevalence study. However, it should be noted that each method would have their own limitations, such as for our work about changing case definitions [3], it could not estimate the number of undetected asymptomatic cases since they could not be identified by any case definitions. For seroprevalence studies, it is possible that some serological responses were caused by cross reaction instead of infections, and therefore the number of undetected cases were overestimated (Please see the discussion for updated manuscript for details). Therefore, we believe our approach is novel in the sense that we provide a novel and valid method to estimate the number of undetected index cases, which could identify asymptomatic cases and without using serology. We still found that a number of cases were undetected as suggested by other methods, which further supported that there are large number of unidentified cases by 'Triangulation'. Triangulation is the practice of obtaining more reliable answers to research questions through integrating results from several different approaches, where each approach has different potential sources of bias that are unrelated to each other [4].

Reviewer #2 (Remarks to the Author):

This manuscript develops a method to improve the estimation of the severity profile of an infectious disease based on secondary cases. This particularly relevant in the context of emerging infectious disease, where information is initially very limited. The method is applied retrospectively to COVID-19.

The paper is relevant and relatively well written. I have a few general comments related to the content and few specific ones related to the form.

Response 2.1: Thank you. We have also updated the review to include articles published by September 2, 2021. We still observe these differences and the conclusion remains the same.

1- Typically, this kind of meta-analysis derives indicators such as combined risk ratios based on mixed effect model. Could the authors derive such pooled estimates from their framework and give indication about the heterogeneity between studies. Typically meta-analyses report such pooled values as well as an I-squared value, which reflects (I believe) the heterogeneity between studies.

Response 2.2: Thank you. We estimated the combined risk ratio by conducting a random effects meta-analysis, and also estimated the pooled value and I-squared value. In general, we found that the heterogeneity for risk ratio comparing the severity of index and secondary cases was low (<75%), except for fever and symptomatic proportion. We added those corresponding results in the revised manuscript. We also added 'We conducted random effects meta-analyses, using the inverse variance method and restricted maximum likelihood estimator for heterogeneity, to summarize the risk ratio among the studies for different severity measures [5-8]. Cochran Q test and the I² statistic were used to identify and quantify heterogeneity among included studies [9, 10]. An I² value more than 75% indicates high heterogeneity [10].' in the method section in the revised manuscript, to clarify the methods we used to determine heterogeneity.

2- While the method seems to give sensible estimates of severity, a lot of new information has been accumulated about COVID-19 severity by now. It seems a missed opportunity not to compare the results from this manuscript to more results from the literature. The submitted manuscript provides a promising method

that could be used in real-time during the next emerging disease event. But retrospectively, a lot of detailed study of COVID-19 have taken place, notably including serological-based evidence of severity profile. So how does the proposed method-estimates compared to what has been published? E.g. more systematic comparison?

Response 2.3: We have updated the review to include articles published by September 2, 2021, and updated the relevant text in the manuscript. The overall conclusions remain the same. Furthermore, we also summarized the systematic reviews we identified in our title screening (Table S7 in revised versions). We have expanded our discussion section to include additional comparisons of our estimates with other reviews. Regarding presence of fever and cough, we found 8 systematic reviews and all have no restrictions on the way of being detected for infected persons, and have higher proportions of fever and cough compared with our estimates, which confirmed the presence of ascertainment bias. We also identified one systematic review reporting 6% CFR for medical personnel.

Regarding the symptomatic proportion of cases, we added serology studies for comparison and we found that the symptomatic proportion based on serology studies could be much lower compared with studies based on PCR-confirmed infections. This difference is also observed in influenza. We have summarized the potential reasons, including 1) the serologic studies could identify infections that missed by PCR, potentially due to imperfect timing and resulting low sensitivity, 2) the estimate from serology could mistakenly classify uninfected persons as infected because of cross reactions in serology, particularly when only convalescent sera are collected.

We added 'Based on the secondary cases, we estimated the proportion of cases with fever and cough were 45% and 36% respectively. Eight previous systematic reviews [11-18] did not stratify their analysis based on the approach for detecting infections (Table S7), and all found that the proportional of fever and cough were higher than our estimates, even though four systematic reviews restricted their analysis to children only, in which the severity of COVID19 tends to be lower compared with adults [14-17], which was also consistent with the ascertainment bias. We estimated 91% of cases were symptomatic,

which was consistent with a previous systematic review suggested that 9% of cases were asymptomatic [19]. However, these estimates could depend on age group, with estimates symptomatic proportion for children was generally lower [14, 15, 17, 19] (Table S7), or converge of test, as the estimates was 73% when restricted to studies with PCR confirmed infection and testing on all contacts. On the other hand, the estimates of symptomatic proportion based on serology data would be much lower, range from 25% and 2-13% for two systematic reviews [20, 21]. This difference is also observed in influenza, where the asymptomatic fraction could be much higher for serology studies compared with case-ascertained studies [22]. One potential reason was the imperfect timing of collection of swabs, resulting in reduced sensitivity to detect infections [23]. Therefore, some infections may be missed by PCR but could be captured by serology [24, 25]. In this case, severity may be overestimated if using PCR-confirmed infections. On the other hand, the estimate from serology could mistakenly classify uninfected persons as infected because of cross reactions in serology, particularly when only convalescent sera are collected [26-30]. Further research on the discrepancy on symptomatic proportion based on serology and PCR would be required.' In the discussion in the revised manuscript.

Where does the figure for MERS-CoV fits in?

Are the figure for death reflecting an CFR or IFR?

Response 2.4: The figure were cited in Discussion section, to illustrate the potential ascertainment bias for MERS. The figure is reflecting CFR (symptomatic case fatality risk), since only symptomatic person would be ascertained in these studies.

Line 83: should read 'by systematically reviewing and analysing'?

Line 109 and throughout: should read 'coverage'?

Line 110: should delet 'except for'?

Line 127: should read 'reaching statistical...'?

Line 149 and other paragraphs below should read 'studies reporting'?

Response 2.5: Thank you for your suggestions and all of them were modified accordingly.

References

1. Maltezou HC, Vorou R, Papadima K, Kossyvakis A, Spanakis N, Gioula G, et al. Transmission dynamics of SARS-CoV-2 within families with children in Greece: A study of 23 clusters. *J Med Virol*. 2021;93(3):1414-20. Epub 2020/08/09. doi: 10.1002/jmv.26394. PubMed PMID: 32767703; PubMed Central PMCID: PMC7441283.
2. Xie W, Chen Z, Wang Q, Song M, Cao Y, Wang L, et al. Infection and disease spectrum in individuals with household exposure to SARS-CoV-2: A family cluster cohort study. *J Med Virol*. 2021;93(5):3033-46. Epub 2021/02/05. doi: 10.1002/jmv.26847. PubMed PMID: 33538342; PubMed Central PMCID: PMC8014049.
3. Tsang TK, Wu P, Lin Y, Lau EHY, Leung GM, Cowling BJ. Effect of changing case definitions for COVID-19 on the epidemic curve and transmission parameters in mainland China: a modelling study. *Lancet Public Health*. 2020;5(5):e289-e96. Epub 2020/04/25. doi: 10.1016/s2468-2667(20)30089-x. PubMed PMID: 32330458; PubMed Central PMCID: PMC7173814.
4. Lawlor DA, Tilling K, Davey Smith G. Triangulation in aetiological epidemiology. *Int J Epidemiol*. 2016;45(6):1866-86. Epub 2017/01/22. doi: 10.1093/ije/dyw314. PubMed PMID: 28108528; PubMed Central PMCID: PMC5841843.
5. Hedges LV, Vevea JL. Fixed-and random-effects models in meta-analysis. *Psychological methods*. 1998;3(4):486.
6. Veroniki AA, Jackson D, Bender R, Kuss O, Langan D, Higgins JPT, et al. Methods to calculate uncertainty in the estimated overall effect size from a random-effects meta-analysis. *Res Synth Methods*. 2019;10(1):23-43. Epub 2018/08/22. doi: 10.1002/jrsm.1319. PubMed PMID: 30129707.
7. Thompson SG, Sharp SJ. Explaining heterogeneity in meta-analysis: a comparison of methods. *Stat Med*. 1999;18(20):2693-708. Epub 1999/10/16. doi: 10.1002/(sici)1097-0258(19991030)18:20<2693::aid-sim235>3.0.co;2-v. PubMed PMID: 10521860.
8. Langan D, Higgins JPT, Jackson D, Bowden J, Veroniki AA, Kontopantelis E, et al. A comparison of heterogeneity variance estimators in simulated random-effects meta-analyses. *Res Synth Methods*. 2019;10(1):83-98. Epub 2018/08/02. doi: 10.1002/jrsm.1316. PubMed PMID: 30067315.

9. Cochran WG. The combination of estimates from different experiments. *Biometrics*. 1954;10(1):101-29.
10. Higgins JP, Thompson SG, Deeks JJ, Altman DG. Measuring inconsistency in meta-analyses. *BMJ*. 2003;327(7414):557-60. Epub 2003/09/06. doi: 10.1136/bmj.327.7414.557. PubMed PMID: 12958120; PubMed Central PMCID: PMC192859.
11. Fathi M, Vakili K, Sayehmiri F, Mohamadkhani A, Hajiesmaeili M, Rezaei-Tavirani M, et al. The prognostic value of comorbidity for the severity of COVID-19: A systematic review and meta-analysis study. *PLoS One*. 2021;16(2):e0246190. Epub 2021/02/17. doi: 10.1371/journal.pone.0246190. PubMed PMID: 33592019; PubMed Central PMCID: PMC7886178.
12. Islam MM, Poly TN, Walther BA, Yang HC, Wang CW, Hsieh WS, et al. Clinical Characteristics and Neonatal Outcomes of Pregnant Patients With COVID-19: A Systematic Review. *Front Med (Lausanne)*. 2020;7:573468. Epub 2021/01/05. doi: 10.3389/fmed.2020.573468. PubMed PMID: 33392213; PubMed Central PMCID: PMC7772992.
13. Wong CKH, Wong JYH, Tang EHM, Au CH, Wai AKC. Clinical presentations, laboratory and radiological findings, and treatments for 11,028 COVID-19 patients: a systematic review and meta-analysis. *Sci Rep*. 2020;10(1):19765. Epub 2020/11/15. doi: 10.1038/s41598-020-74988-9. PubMed PMID: 33188232; PubMed Central PMCID: PMC7666204.
14. Chang TH, Wu JL, Chang LY. Clinical characteristics and diagnostic challenges of pediatric COVID-19: A systematic review and meta-analysis. *J Formos Med Assoc*. 2020;119(5):982-9. Epub 2020/04/21. doi: 10.1016/j.jfma.2020.04.007. PubMed PMID: 32307322; PubMed Central PMCID: PMC7161491.
15. Li B, Zhang S, Zhang R, Chen X, Wang Y, Zhu C. Epidemiological and Clinical Characteristics of COVID-19 in Children: A Systematic Review and Meta-Analysis. *Front Pediatr*. 2020;8:591132. Epub 2020/11/24. doi: 10.3389/fped.2020.591132. PubMed PMID: 33224909; PubMed Central PMCID: PMC7667131.

16. Panahi L, Amiri M, Pouy S. Clinical Characteristics of COVID-19 Infection in Newborns and Pediatrics: A Systematic Review. *Arch Acad Emerg Med.* 2020;8(1):e50. Epub 2020/05/23. PubMed PMID: 32440661; PubMed Central PMCID: PMC7212072.
17. Cui X, Zhao Z, Zhang T, Guo W, Guo W, Zheng J, et al. A systematic review and meta-analysis of children with coronavirus disease 2019 (COVID-19). *J Med Virol.* 2021;93(2):1057-69. Epub 2020/08/08. doi: 10.1002/jmv.26398. PubMed PMID: 32761898; PubMed Central PMCID: PMC7436402.
18. Hashan MR, Smoll N, King C, Ockenden-Muldoon H, Walker J, Wattiaux A, et al. Epidemiology and clinical features of COVID-19 outbreaks in aged care facilities: A systematic review and meta-analysis. *EClinicalMedicine.* 2021;33:100771. Epub 2021/03/09. doi: 10.1016/j.eclinm.2021.100771. PubMed PMID: 33681730; PubMed Central PMCID: PMC7917447.
19. He J, Guo Y, Mao R, Zhang J. Proportion of asymptomatic coronavirus disease 2019: A systematic review and meta-analysis. *J Med Virol.* 2021;93(2):820-30. Epub 2020/07/22. doi: 10.1002/jmv.26326. PubMed PMID: 32691881; PubMed Central PMCID: PMC7404334.
20. Oran DP, Topol EJ. The Proportion of SARS-CoV-2 Infections That Are Asymptomatic : A Systematic Review. *Ann Intern Med.* 2021;174(5):655-62. Epub 2021/01/23. doi: 10.7326/M20-6976. PubMed PMID: 33481642; PubMed Central PMCID: PMC7839426.
21. Chen X, Chen Z, Azman AS, Deng X, Sun R, Zhao Z, et al. Serological evidence of human infection with SARS-CoV-2: a systematic review and meta-analysis. *Lancet Glob Health.* 2021;9(5):e598-e609. Epub 2021/03/12. doi: 10.1016/s2214-109x(21)00026-7. PubMed PMID: 33705690; PubMed Central PMCID: PMC8049592.
22. Leung NH, Xu C, Ip DK, Cowling BJ. The Fraction of Influenza Virus Infections That Are Asymptomatic: A Systematic Review and Meta-analysis. *Epidemiology.* 2015. doi: 10.1097/EDE.0000000000000340. PubMed PMID: 26133025.
23. Kucirka LM, Lauer SA, Laeyendecker O, Boon D, Lessler J. Variation in False-Negative Rate of Reverse Transcriptase Polymerase Chain Reaction-Based SARS-CoV-2 Tests by Time Since Exposure. *Ann Intern Med.* 2020;173(4):262-7. Epub 2020/05/19. doi: 10.7326/M20-1495. PubMed PMID: 32422057; PubMed Central PMCID: PMC7240870.

24. Miller TE, Garcia Beltran WF, Bard AZ, Gogakos T, Anahtar MN, Astudillo MG, et al. Clinical sensitivity and interpretation of PCR and serological COVID-19 diagnostics for patients presenting to the hospital. *FASEB J.* 2020;34(10):13877-84. Epub 2020/08/29. doi: 10.1096/fj.202001700RR. PubMed PMID: 32856766; PubMed Central PMCID: PMC7461169.
25. Long QX, Liu BZ, Deng HJ, Wu GC, Deng K, Chen YK, et al. Antibody responses to SARS-CoV-2 in patients with COVID-19. *Nat Med.* 2020;26(6):845-8. Epub 2020/05/01. doi: 10.1038/s41591-020-0897-1. PubMed PMID: 32350462.
26. Shrock E, Fujimura E, Kula T, Timms RT, Lee IH, Leng Y, et al. Viral epitope profiling of COVID-19 patients reveals cross-reactivity and correlates of severity. *Science.* 2020;370(6520). Epub 2020/10/01. doi: 10.1126/science.abd4250. PubMed PMID: 32994364; PubMed Central PMCID: PMC7857405.
27. Tso FY, Lidenge SJ, Pena PB, Clegg AA, Ngowi JR, Mwaiselage J, et al. High prevalence of pre-existing serological cross-reactivity against severe acute respiratory syndrome coronavirus-2 (SARS-CoV-2) in sub-Saharan Africa. *Int J Infect Dis.* 2021;102:577-83. Epub 2020/11/12. doi: 10.1016/j.ijid.2020.10.104. PubMed PMID: 33176202; PubMed Central PMCID: PMC7648883.
28. Lustig Y, Keler S, Kolodny R, Ben-Tal N, Atias-Varon D, Shlush E, et al. Potential antigenic cross-reactivity between SARS-CoV-2 and Dengue viruses. *Clin Infect Dis.* 2020. Epub 2020/08/17. doi: 10.1093/cid/ciaa1207. PubMed PMID: 32797228; PubMed Central PMCID: PMC7454334.
29. Hicks J, Klumpp-Thomas C, Kalish H, Shunmugavel A, Mehalko J, Denson JP, et al. Serologic Cross-Reactivity of SARS-CoV-2 with Endemic and Seasonal Betacoronaviruses. *J Clin Immunol.* 2021;41(5):906-13. Epub 2021/03/17. doi: 10.1007/s10875-021-00997-6. PubMed PMID: 33725211; PubMed Central PMCID: PMC7962425.
30. Huang AT, Garcia-Carreras B, Hitchings MDT, Yang B, Katzelnick LC, Rattigan SM, et al. A systematic review of antibody mediated immunity to coronaviruses: kinetics, correlates of protection, and association with severity. *Nat Commun.* 2020;11(1):4704. Epub 2020/09/19. doi: 10.1038/s41467-020-18450-4. PubMed PMID: 32943637; PubMed Central PMCID: PMC7499300.

REVIEWERS' COMMENTS

Reviewer #1 (Remarks to the Author):

Thanks for the revised manuscript. While I think their literature review offers some new insights, I am afraid I am still not convinced that the statistical method developed added much novelty and values to the literature. For example, many SEIR type modelling approaches (e.g. Li. et al, Science, 2020) can naturally incorporate and estimate asymptomatic class without relying on serology data.

Reviewer #2 (Remarks to the Author):

I am happy with authors' modifications, and I support the publication of the manuscript in nature comms.

The only two comments I have are:

- In the SI, I believe something went wrong the references format. Once fixed, I also suggest to submit the SI as a pdf to avoid further issues.
- In the first paragraph of the discussion, you state the 'ascertainment bias exists among studies'. This is a bit vague, could you not state that the bias was actually consistent across studies given the relatively low heterogeneity?

REVIEWERS' COMMENTS

Reviewer #1 (Remarks to the Author):

Thanks for the revised manuscript. While I think their literature review offers some new insights, I am afraid I am still not convinced that the statistical method developed added much novelty and values to the literature. For example, many SEIR type modelling approaches (e.g. Li. et al, Science, 2020) can naturally incorporate and estimate asymptomatic class without relying on serology data.

Response 1.1: Thank you for the reviewer's support on our literature review. While we agree that the SEIR-type modelling approaches could also estimates asymptomatic cases, additional data must still be added to do so. For the example (Li. et al, Science, 2020) mentioned by the reviewer, it used the mobility data to conduct such inferences, which also relied on the validity of the mobility data. As mentioned by previous responses, we believe our work could provide additional evidence to support the large number of unidentified cases by Triangulation, a practice of obtaining more reliable answers to research questions through integrating results from several different approaches, where each approach has different potential sources of bias that are unrelated to each other.

Reviewer #2 (Remarks to the Author):

I am happy with authors' modifications, and I support the publication of the manuscript in nature comms.

The only two comments I have are:

- In the SI, I believe something went wrong the references format. Once fixed, I also suggest to submit the SI as a pdf to avoid further issues.

Response 2.1: We apologize such issues in the references. We fixed this and also submitted this as a pdf file.

- In the first paragraph of the discussion, you state the ‘ascertainment bias exists among studies’. This is a bit vague, could you not state that the bias was actually consistent across studies given the relatively low heterogeneity?

Response 2.2: We agree with reviewer’s suggestion, and we modified this accordingly. We modified this phase to ‘ascertainment bias was consistent across studies’.